# Asymmetric Coordination Environment Engineering of Atomic Catalysts for CO_2_ Reduction

**DOI:** 10.3390/nano13020309

**Published:** 2023-01-11

**Authors:** Xianghua Hou, Junyang Ding, Wenxian Liu, Shusheng Zhang, Jun Luo, Xijun Liu

**Affiliations:** 1Center for Electron Microscopy and Tianjin Key Lab of Advanced Functional Porous Materials, Institute for New Energy Materials & Low-Carbon Technologies, School of Materials, Tianjin University of Technology, Tianjin 300384, China; 2MOE Key Laboratory of New Processing Technology for Non-Ferrous Metals and Materials, Guangxi Key Laboratory of Processing for Non-Ferrous Metals and Featured Materials, School of Resource, Environments and Materials, Nanning 530004, China; 3College of Materials Science and Engineering, Zhejiang University of Technology, Hangzhou 310014, China; 4College of Chemistry, Zhengzhou University, Zhengzhou 450001, China

**Keywords:** asymmetric atom sites, coordination structure, carbon dioxide reduction reaction, catalyst design

## Abstract

Single-atom catalysts (SACs) have emerged as well-known catalysts in renewable energy storage and conversion systems. Several supports have been developed for stabilizing single-atom catalytic sites, e.g., organic-, metal-, and carbonaceous matrices. Noticeably, the metal species and their local atomic coordination environments have a strong influence on the electrocatalytic capabilities of metal atom active centers. In particular, asymmetric atom electrocatalysts exhibit unique properties and an unexpected carbon dioxide reduction reaction (CO_2_RR) performance different from those of traditional metal-N_4_ sites. This review summarizes the recent development of asymmetric atom sites for the CO_2_RR with emphasis on the coordination structure regulation strategies and their effects on CO_2_RR performance. Ultimately, several scientific possibilities are proffered with the aim of further expanding and deepening the advancement of asymmetric atom electrocatalysts for the CO_2_RR.

## 1. Introduction

Since the first industrial revolution, the massive consumption of fossil fuels has promoted the innovation of modern science and technology and greatly raised the quality of human lifestyles [1,2,3,4,5]. However, the challenges of energy scarcity and atmospheric pollution are in irreversible progress, and it is imperative to formulate a clean and sustainable energy technology route [6,7,8,9,10]. In particular, one of the possible alternatives is to employ electrocatalytic conversions reactions to manufacture clean energy from renewable or sustainable green sources, e.g., the hydrogen evolution reaction (HER) [11,12], oxygen evolution reaction (OER) [13,14], oxygen reduction reaction (ORR) [15,16], CO_2_ reduction reaction (CO_2_RR) [17,18], nitrogen reduction reaction (NRR) [19,20], nitrate reduction reaction (NO_3_RR), etc. [21,22,23]. In these proposed electrochemical reactions, the introduction of catalysts significantly reduces the extra electric energy consumption [24,25,26,27,28]. For traditional bulk and/or nanoparticles (NPs) electrocatalysts, only a few layers of the atoms on the surfaces of the catalysts are involved in the reactions, resulting in a large percentage of the metal atoms being ineffective and wasted. Consequently, next-generation electrocatalysts need to meet the strict demands of higher atomic utilization efficiency compared with the original ones [29,30,31,32].

In 2011, Zhang et al. prepared an atomically dispersed Pt_1_/FeO_x_, which was the first time the concept of a “single-atom catalyst” (SAC) had ever been applied worldwide [33]. In recent years, due to their almost 100% atomic utilization rates and exceptional activity and selectivity, single-atom electrocatalysts have gained significant scientific attention and their development has been actively underway [34,35,36]. First, SACs loaded on supports not only have relatively uniform and well-defined active centers with modulated coordination bonds similar to homogeneous catalysts but also retain the advantages of easy separation, reusability, and higher stability of non-homogeneous catalysts. Thus, SACs bridge the gap between the two as a result [37,38,39,40]. Furthermore, advanced characterization strategies, as well as density functional theory (DFT) simulations, provide significant ways to comprehend the relationship between the atomic coordination environment and reaction mechanisms, which benefits from the fact that the simple electronic and coordination structures of SACs make it straightforward to identify and accurately define their functional components [41,42,43]. Moreover, recognizing the connection between the isolated active atomic sites and catalytic properties enables us to better define metal atom types, coordination numbers, nearby coordination dopants, geometry shapes, and electronic structures, and this can also substantially enhance the intrinsic comprehension of their activity and selectivity [44,45].

Noticeably, as shown in Figure 1, the coordination environment is as crucial as the metal center, which not only stabilizes the metal atoms but also determines the activity, selectivity, and stability of SACs. In general, the electronic structure and/or oxidation state of the metal center can be directly impacted by the first-shell atoms (direct coordination), including the type, number, and axial chemical environment of the coordination atoms, while the second- or higher-shell environment (indirect coordination) can somewhat influence the CO_2_RR’s performance. The classical symmetric planar four-coordination structure (MN_4_: M, metal atoms; N, nitrogen atom) is one of the common structures of the active centers of carbon-based SACs for the CO_2_RR. However, there are restrictions to symmetric MN_4_ monoatomic sites in electrocatalytic reactions, i.e., the inappropriate adsorption strength with intermediates might result in the symmetric coordination of N atoms with high electronegativity, which causes the whole reaction process to lag. This challenge, by modifying the local coordination environment of the central metal atom, i.e., introducing the corresponding asymmetric coordination structure, can be solved to some extent. For instance, breaking the original M-N bond of MN_4_ can lead to a low-coordination site MNx (x ≤ 3) with vacancy defects. Alternatively, by introducing heteroatoms to replace some of the coordination N atoms of MN_4_, it becomes possible to construct MNxY with heteroatomic coordination structure sites (Y is other coordination atoms such as O, S, Cl, F, etc.), and the heteroatoms can also replace atoms on different coordination shells around the metal center. In addition, when two isolated MN_4_ sites are close enough, they can transform into neighboring metal atom sites M_1_M_2_Nx. These asymmetric atomic sites, by adjusting the electronic structure of the catalyst to break the inappropriate adsorption strength between the atomic active sites and the reaction intermediates, optimize the adsorption-desorption process of reactants and products and further improve the electrocatalytic performance, demonstrating how the local coordination environment has a significant impact on the energy barriers and reaction paths [46,47,48,49,50].

So far, many impressive review papers have summarized the synthesis, characterization, and catalytic performance of SACs [51,52,53]. However, methods for breaking the symmetry of metal-N_4_ sites, and the unique electronic structure and CO_2_RR catalytic behavior of asymmetric atom sites, are rarely summarized [54,55]. In this review, we introduce the influence of the asymmetric coordination structure on the intrinsic electrochemical activity of SACs for the CO_2_RR, including low coordination, heteroatom coordination, dual-metal sites, and the second coordination shell. In addition, the synthesis strategy of asymmetric SACs, and the structure-function relationship between coordination structure and CO_2_RR performance are highlighted (Figure 1). Finally, the challenges, opportunities, and future development directions of SACs for electrocatalytic energy conversion technology are discussed.

**Figure 1 nanomaterials-13-00309-f001:**
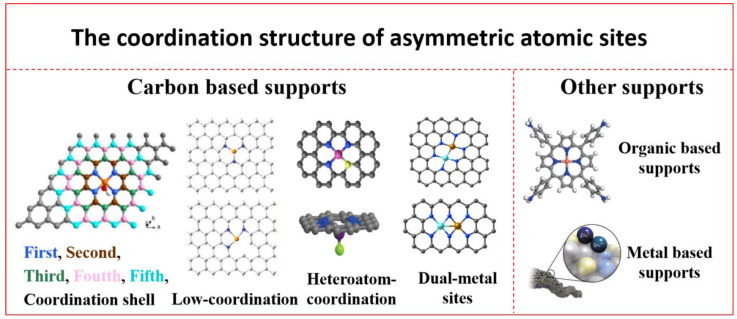
Model diagrams of symmetric and asymmetric atomic configurations. (The heteroatom coordination (bottom) elements are reprinted with permission from ref. [56], copyright: (2019) Nature Publishing Group; the organic-based support elements are reprinted with permission from ref. [57], copyright: (2021) Nature Publishing Group; other elements are taken from Figure 2, Figure 3, Figure 4, Figure 5, Figure 6 and Figure 7.)

## 2. Advantages of Asymmetrically Coordinated SACs

Initially, in 2015, Varela et al. reported that the carbon-based catalysts (Fe/Mn-N-C) with doped N and Fe/Mn species demonstrated a higher CO selectivity of up to 80% at −0.5 V vs. RHE, which is superior to polycrystalline Au electrodes [58] and other comparable metal-N-C materials (e.g., the Co protoporphyrin, Ni^2+^ on N-doped graphene, Rh-porphyrin-like functionalized graphene, etc.) [59,60,61]. Then, the researchers identified that FeN_4_ moieties were the primary active sites for the selective CO_2_ reduction into CO with up to 80% FE_CO_ in an aqueous solution (Figure 2a,c) [62], whereas Fe nanoparticles (NPs) mostly catalyzed hydrogen evolution [63]. Through using DFT calculations, the CO_2_RR mechanism of Fe-N_4_-embedded N-doped graphene was further investigated, finding that the CO_2_ molecules preferred to adsorb on the Fe catalytic site, while the neighboring N atoms promoted the formation of COOH* and the release of CO*, thus improving the conversion efficiency (Figure 2b) [62]. 

**Figure 2 nanomaterials-13-00309-f002:**
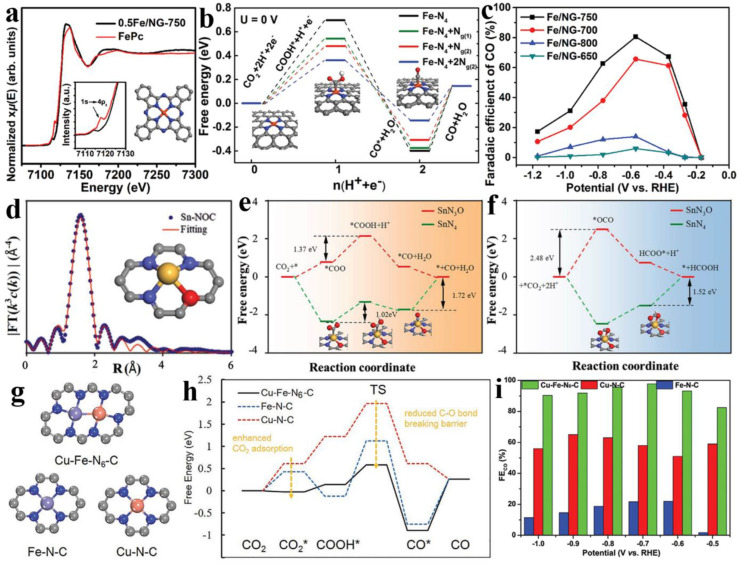
(**a**) Normalized Fe K-edge X-ray adsorption near-edge structure (XANES) spectra of 0.5Fe/NG-750 catalyst. (**b**) Free energy diagram for electrochemical CO_2_ reduction to CO. (**c**) CO Faradaic efficiency (FE_CO_). Reprinted with permission from ref. [62]. Copyright: (2018) WILEY-VCH Verlag GmbH & Co. KGaA, Weinheim. (**d**) The Fourier transform extended X-ray absorption fine structure (FT-EXAFS) fitting curve for Sn-NOC in R space. The calculated Gibbs free energy diagrams for (**e**) CO_2_-to-CO conversion and for (**f**) CO_2_-to-HCOOH conversion. Reprinted with permission from ref. [64]. Copyright: (2021) The Authors. Advanced Science published by Wiley-VCH GmbH. (**g**) Atomic models of Cu-Fe-N_6_, Fe-N_4_, and Cu-N_4_ catalytic centers. (**h**) Free energy profiles for CO_2_ electroreduction reactions. (**i**) FE_CO_ of Fe-N-C, Cu-N-C, and Cu-Fe-N_6_-C. Reprinted with permission from ref. [65]. Copyright: (2020) Wiley-VCH GmbH.

Furthermore, in a work on a N-doped porous carbon matrix including isolated FeN_x_ moieties with different N coordination numbers, it was revealed that when compared with perfect FeN_4_ and the Fe-N_3_ atomic site, the low-coordinated FeN_3_V active center effectively reduced the free energy barrier for promoting the formation of intermediates *COOH of rate determination step, so that Fe_1_NC/S_1_-1000 with Fe-N_3_V atomic sites demonstrated a high 96% FE_CO_ [66]. In addition to controlling the N coordination number, heteroatoms exist in different coordination shell locations or spatial orientations to alter the catalytic activity of the atomic metal center. For instance, in contrast with the traditional Sn-N_4_ configuration, which mostly produces HCOOH and H_2_ products, N-rich carbon catalysts made up of atomically dispersed SnN_3_O_1_ active sites have CO as the main product (Figure 2d). According to DFT calculations, SnN_3_O_1_′s atomic configuration lowers the activation energy barrier for *COO and *COOH generation (△G = 1.37 eV) while significantly raising the energy barrier for the production of *OCO (△G = 2.48 eV), which encourages the conversion of CO_2_ to CO and inhibits the production of HCOOH, opening up a new route for the selectivity of particular products (Figure 2e,f) [64]. Single atoms can also form metal-metal bonds with other metals in order to build binary atom sites with the unique advantage of synergistic effects. The resulting Fe/Cu-N-C catalysts exhibited excellent >95% FE_CO_ with a wide potential range from −0.4 to −1.1 V vs. RHE (Figure 2g,i), which was primarily attributed to the bifunctional active sites, i.e., the Cu site has a greater affinity for molecular CO_2_ (CO_2_ adsorption energy: Cu-Fe-N_6_-C, −0.03 eV; Fe-N-C, 0.43 eV; Cu-N-C, 0.61 eV), increasing the CO_2_ concentration, while the Fe site inhibits hydrogen evolution (Figure 2h) [65,67]. In conclusion, through the modification of the MN_4_ coordination environment, including the metal center, the coordination atom numbers, the coordination atom species, and the position or spatial orientation of the coordination shell had a significant influence on the microscopic properties and catalytic activity of the SACs. Based on a more detailed comparative evaluation, owing to the high electronegative N components in the local structure, the symmetric MN_4_ moiety may lead to an inappropriate adsorption strength with the intermediate and can slow down the entire reaction process, causing problems such as low product selectivity and a high reaction overpotential. Meanwhile, the active site’s electronic and geometric asymmetry can control the adsorption-desorption behavior of the crucial reaction intermediates, which is beneficial in lowering the reaction energy barrier, preventing the occurrence of side reactions, and accelerating the reaction rate, all of which contribute to improved catalytic activity.

## 3. The Synthesis Strategies for Breaking MN_4_

There are two main routes to breaking the M-N bond of MN_4_. One is the carbonization of precursors such as metal-organic frameworks (MOFs) and polymers with nitrogen-containing specie ligands or other mixtures containing metal salt and nitrogen sources (such as urea, melamine, dicyandiamide, N_2_, NH_3_, etc.), in which the coordination number can be controlled by finely modulating temperature [68]. Another is the pre-construction of N-doped graphene with abundant defects, vacancies, or edges, and subsequently introducing metal sources that are immobilized in the pre-existing positions of graphene via pyrolytic dispersion to produce low-coordinated metal atoms [69,70]. For instance, the weaker M-O bond is broken in the M-N_x_O_y_ hybrid coordination structure, followed by the removal of the oxygen atom, producing an atomically scattered M-N_x_ catalyst [71]. In addition, M-N bonds can also be broken by using sophisticated plasma treatments [72,73].

The introduction of coordinated heteroatoms beyond N can be achieved via the following methods: (i) pre-storing in the precursor through coordination, impregnation, encapsulation, grinding, or other physical mixing styles; (ii) using biomass precursors or polymers rich in a variety of non-metallic elements (purslane, chitosan, cyanobacteria, etc.); (iii) for materials with low boiling points, such as S, P, and I, one can diffuse onto the carbon substrate via the sublimation effect during carbonization, coordinating with metals, or replacing carbon atoms; and (iv) using reactants that contain certain heteroatoms, such as S powder, thiourea, and sulfur salts for S, triphenylphosphine, NaH_2_PO_2_, and polypyrrole for P, NH_4_I and I_2_ for I, HCl, NaCl, and KCl for Cl, borane and boronic acid for B, and polytetrafluoroethylene for F [74,75,76,77,78,79,80,81,82,83]. 

The construction of homonuclear/heteronuclear bimetallic atoms is commonly achieved via increasing the concentration or species of the metal salts, usually accompanied by monometallic sites. For the fabrication of homonuclear sites, the increased metal amount means raising the possibility of NPs [84]. Ligand-protected diatomic nodes can be designed using MOF substrates acting as metal precursors [85]. Recently, an anion replacement deposition-precipitation method was reported, in which water-soluble M_2_ precursors become insoluble after anion replacement and were thus deposited onto a pre-prepared carrier [86]. For heteronuclear sites, one method is to encapsulate two metal salts into the pore channels of the MOF template [87], and another is to prepare a mixed precursor containing a nitrogen source, carbon source, and double metallic source [88]. Preparing high-quality bimetallic structures using atomic layer deposition (ALD) technologies, where the second metal can only be attached to the primary metal by modulating the deposition conditions, has also been demonstrated to be effective [89].

## 4. The Characterization of Asymmetric Atom Sites

Currently, there are two main characterization tools for describing single atomic sites: high-angle annular dark-field scanning transmission electron microscopy (HAADF-STEM) and XAFS. 

The resolution of HADDF-STEM images can reach up to 1 Å, allowing a fine distinction between individual atoms [90]. For instance, Zhang et al. reported a conception in which graphene defects capture Ni single atoms as active sites, as shown in Figure 3a, and identified the presence of di-vacancy defects that trap an atomic Ni species (aNi@Di-vacancy) [91]. Furthermore, Liu et al. synthesized an atomic dispersed cobalt-based catalyst with a doped I species that has low-coordination CoN_3_ and IN_3_ sites (Co-I-N/G). Through atomic-scale HAADF-STEM images (Figure 3b) and a Z-contrast analysis, two sets of uniformly distributed bright spots with different intensities (marked by green and yellow circles, respectively) were attributed to Co and I atoms, respectively [92]. For dual-atom sites, Zhou et al. presented a bimetallic (Rh-Fe) interbond as an ideal chemical facilitator (FR-NCS) that can facilitate the dispersion of Fe atoms. In Figure 3c, most of the Fe and Rh are uniformly and atomically dispersed on the nitrogen-doped carbon hollow spheres shell (highlighted by blue circles), and Fe-Rh “hetero-pair” atoms are found in Figure 3d, demonstrating the formation of an Rh-Fe interband [93]. In short, it can be noted that the STEM technique does not acquire accurate structural information about the atom species and coordination number. 

EXAFS can be used to calibrate the atomic species and obtain the neighboring atom structure, and, thus, is an indispensable tool for investigating asymmetric coordination [94]. Sui et al. synthesized an effective CO_2_RR catalyst featuring Ag atoms coordinated with three nitrogen atoms (Ag_1_-N_3_, Figure 3e). In the EXAFS spectra, a significant peak in the Ag-SACs’ EXAFS spectra at 1.5 Å is attributed to Ag-N scattering, but there is no Ag-Ag scattering at 2.6 Å, indicating an atomically dispersed Ag species (Figure 3f). In the EXAFS fitting curves (Figure 3g,h), the Ag-N coordination numbers of Ag_1_-N_3_/PCNC and Ag_1_-N_2_/PCNC are fitted from 3.1 to 1.9, respectively, demonstrating the formation of Ag_1_-N_3_ and Ag_1_-N_2_ coordination structures. The electrochemical test results showed the trend of the CO Faradaic efficiency (FE_CO_) sequence, Ag_1_-N_3_/PCNC > Ag_1_-N_2_/PCNC > Ag NPs/C > CFP, confirming that Ag_1_-N_3_ was the best active site among them (Figure 3i) [95].

**Figure 3 nanomaterials-13-00309-f003:**
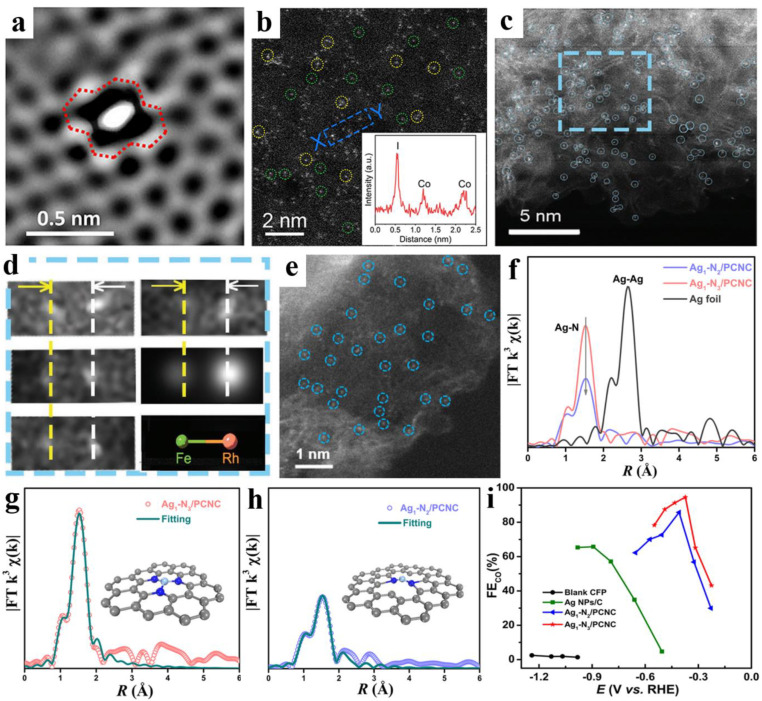
(**a**) HAADF-STEM image of the defective area with atomic trapped Ni. Reprinted with permission from ref. [91]. Copyright: (2017) Elsevier Inc. (**b**) Atomic-scale HAADF-STEM image of Co-I-N/G. Reprinted with permission from ref. [92]. Copyright: (2021) American Chemical Society. (**c**) HAADF-STEM image of FR-NCS. (**d**) The zoomed-out blue rectangular area showing the existence of adjacent dual-metal atoms. Reprinted with permission from ref. [93]. Copyright: (2020) Wiley-VCH GmbH. (**e**) Aberration-corrected STEM images of Ag_1_-N_3_/PCNC. (**f**) EXAFS spectra. (**g**,**h**) EXAFS fitting curves at R space. (**i**) FE_CO_. Reprinted with permission from ref. [95]. Copyright: (2021) American Chemical Society.

## 5. Asymmetric Atom Sites for CO_2_RR

### 5.1. Low-Coordination Structure

Eliminating part of the coordinated N atoms from the original symmetric MN_4_ results in the corresponding defective sites, i.e., the low-coordinated MNx configuration. The low-coordination sites of MN_x_ (M = Fe, Co, Ni, Cu, Zn, Mn, Ag, etc.; x < 4) are mainly constructed in two strategies: one is a one-step method to obtain the desired coordination numbers, and the other is breaking the original metal-coordination bond.

In abundant atomically dispersed based metal-nitrogen co-decorated carbon (M-NC) materials, some transition metals, i.e., Fe, Co, Ni, and Cu, can facilitate charge transfer to account for their unfilled 3d orbitals. Wang et al. reported the gas diffusion of ferrocene and thermal activation (800–1000 °C) to successfully fabricate isolated Fe atoms loaded on N-doped porous carbon polyhedrons (Figure 4a) with different N coordination numbers (FeN_4_, FeN_3_). According to DFT calculations and experimental measurements, the Fe-N_3_V active centers have most balanced free energy barriers of intermediates, CO_2_ to *COOH (0.47 eV), *COOH to *CO (0.15 eV), and *CO to CO (0.14 eV), strengthening its CO_2_RR performance. At the same applied potential of −0.5 V vs. RHE, the FE_CO_ of Fe_1_NC/S_1_-1000, Fe_1_NC/S_1_-900, and Fe_1_NC/S_1_-800 are 96%, 90%, and 82%, respectively (Figure 4b,c) [66]. Similarly, three atomically dispersed Co catalysts (800 °C, Co-N_4_; 900 °C, Co-N_3_; and 1000 °C, Co-N_2_) can be selectively synthesized by adopting various pyrolysis temperatures of bimetallic Co/Zn ZIFs. The Co-N_2_ could acquire a better selectivity and activity compared with other Co-N_x_, which demonstrated that the inert reactive sites could also be activated by adjusting the coordination environment (Figure 4d,e) [96]. Rong et al. adopted the other strategy that differs from those mentioned above to construct a vacancy-defect Ni site. The single-atom Ni-N_3_O mixed-coordination precursors were first synthesized at 500 °C and then heated to 800 °C, where the oxygen atoms could be subsequently eliminated on account of the weaker Ni-O interactions, leading to atomically dispersed Ni-N_3_-V catalysts (Figure 4f). It was discovered that the electrocatalytic activity of the CO_2_RR can be significantly enhanced after the introduction of vacancy defects in Ni-N_3_-V SACs, compared with the difficulty of the NiN_3_ site to release CO (△G = 1.264 eV), and this is mainly attributed to Ni-N_3_-V’s optimized and moderate CO_2_ to COOH* reaction free energy (△G = 0.680 eV) (Figure 4g) [71]. Calculations further disclose that the active site is the particular defect-Ni-N_3_ structure [97]. Moreover, the Ni-pyridinic N_2_V_2_ is quickly generated as a result of the high-energy plasma’s constant collision and presents a more advantageous comprehensive performance than other Ni-pyridinic N_x_C_3−x_ species, resulting in a higher CO_2_RR efficiency and worse HER activity (Figure 4h,i) [72]. Lin et al. found that the FE_CO_ of both the NiN_2_ site and FeN_4_ site is proportional to the reaction temperature, but that the NiN_2_ site is more effective, which stems from the variation in the adsorption strength of the key intermediates between the two [98]. Cu-based nanocatalysts are frequently utilized for the construction of a wide range of C_1+_ and C_2+_ products [99]. The work of Guan et al. showed the significance of the distance between isolated atomic copper sites for the electroreduction of CO_2_ to various hydrocarbons (Figure 4j). DFT calculations indicate that high 4.9%_mol_ Cu concentrations with adjacent Cu-N_2_ sites are more advantageous for the formation of C_2_H_4_ due to the reduced free energy of the C-C coupling, while all isolated Cu-N_4_, neighboring Cu-N_4_, and isolated Cu-N_2_ sites tend to generate C_1_ products when the concentration of Cu is less than 2.4%_mol_ [100]. 

**Figure 4 nanomaterials-13-00309-f004:**
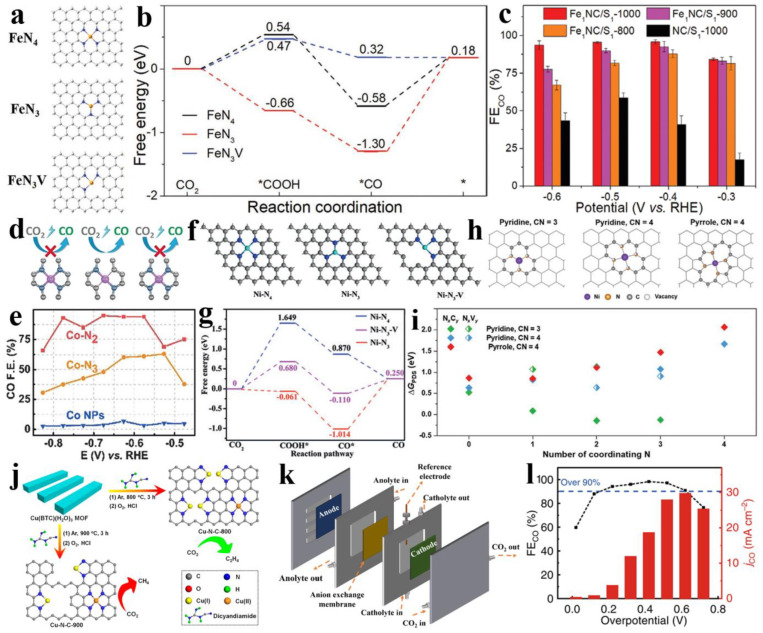
(**a**) The optimized atomic structures of FeN_4_, FeN_3_, and FeN_3_V. (**b**) The calculated free-energy diagrams. (**c**) FE_CO_. Reprinted with permission from ref. [66]. Copyright: (2020) WILEY-VCH Verlag GmbH & Co. KGaA, Weinheim. (**d**) Atomic structures of CoN_4_ and CoN_2_ on graphene. (**e**) FE_CO_. Reprinted with permission from ref. [96]. Copyright: (2018) Wiley-VCH Verlag GmbH & Co. KGaA, Weinheim. (**f**) The optimized atomic structures of Ni SACs. (**g**) Calculated free-energy diagram. Reprinted with permission from ref. [71]. Copyright: (2020) Wiley-VCH Verlag GmbH & Co. KGaA, Weinheim. (**h**) Different coordination structures of NiNG SACs. (**i**) The calculated free energy change at potential determining step (PDS; ΔGPDS). Reprinted with permission from ref. [72]. Copyright: (2021) Wiley-VCH GmbH. (**j**) Schematic of preparation of the Cu-N-C-T catalysts. Reprinted with permission from ref. [100]. Copyright: (2020) American Chemical Society. (**k**) Schematic of flow-cell configuration. Reprinted with permission from ref. [68]. Copyright: (2021) Wiley-VCH GmbH. (**l**) FE_CO_ and current density of CO (j_CO_) of Mn-C_3_N_4_/CNT in the CO_2_-saturated [Bmim]BF_4_/CH_3_CN-H_2_O electrolyte. Reprinted with permission from ref. [101]. Copyright: (2020) Nature Publishing Group.

To date, Zn-NC has received less attention compared with other transition metals (Fe, Co, Ni, and Cu), and it is generally believed that when the 3d^10^ orbital is fully occupied, it can significantly impair electron transport and lower the electrocatalytic activity of Zn. Li et al. reported a N-anchored low-valence Zn single-atom catalyst that contains saturated four-coordinate (ZnN_4_) and unsaturated (Zn-N_3_) active sites. The unsaturated Zn-N_3_ could dramatically reduce the energy barrier by stabilizing the COOH* intermediate, thus allowing simultaneous large current densities of up to 1 A cm^−2^ and high CO selectivity of up to 95% using flow cell devices (Figure 4k) [68]. Compared with the poor selectivity and activity (65% FE_CO_, 3.3 mA cm^−2^ of j_CO_) of Mn-N_4_ [58], the Mn-N_3_ center in Mn-C_3_N_4_/CNT can attain an excellent CO_2_RR performance (98.8% FE_CO_, 14.0 mA cm^−2^ of j_CO_). In situ X-ray absorption spectra and DFT calculations reveal that the Mn-N_3_ site can lower the free energy barrier for the production of the crucial intermediate COOH*, which is responsible for high catalytic performance (Figure 4l) [101]. Additionally, altering the coordinate environment of the Ag atoms can boost their activity. It was shown that the Ag_1_-N_3_ site performed better than Ag_1_-N_2_ and Ag NPs, owing to its lower CO binding energy [95]. 

The investigation of the above works reveals that the reduced oxidation state of the monoatomic metal center with low coordination numbers makes it easier to adsorb reactant molecules, and optimize the adsorption strength of intermediates, especially COOH* in the CO product process. In addition, we also speculate that the extra space caused by low coordination also facilitates a more flexible matching between reactant molecules and active sites.

### 5.2. Lateral Heteroatom Coordination Structure

The doping of non-metallic atoms in carbon supports is an efficient and flexible technique for improving the overall performance of SACs, such as through the introduction of heteroatoms to replace the coordinated N atoms in the symmetrical planar MN_4_ sites to form a lateral heteroatom coordination structure. The laterally coordinated non-metallic heteroatoms, such as C, N, S, O, B, P, F, etc., could modulate the coordination environment of the atomic metal active center through direct coordination (first coordination shell) and indirect coordination (second or higher coordination shell), thereby achieving the goal of optimizing the reaction path.

In 2017, Zhao et al. reported ZIF-8-derived NiN_3_C_1_ single atomic catalysts for the efficient electroreduction of CO_2_. The positive correlation between HCO^3−^ concentration and the efficiency of CO_2_ electroreduction was disclosed (Figure 5a), which mainly benefits from the dual roles of HCO^3−^ (the proton donor in the reaction solution, the ideal equilibrium carrier for CO_2_) [102]. Gong et al. used non-nitrogenous MOFs, bimetallic MgNi-MOF-74 with extra pyrrole (Py) N sources, to construct a series of NiSA-N_x_-C (x = 2, 3, 4) at various carbonation temperatures (Figure 5b), wherein the NiSA-N_2_-C could easily facilitate the formation of the COOH* intermediate and therefore result in its remarkable activity (Figure 5c) [103]. Yang et al. also observed that the N coordination number dropped from 4 to 1, while the C coordination number grew throughout the construction process of the Ni@N_x_C_y_ catalysts when the temperature was raised from 800 to 1100 °C [104]. Of note is the charge capacity of the site for facilitating the electrochemical steps as well as hydrogen bonding to the intermediate for stabilizing the intermediates, which were often ignored in previous DFT calculations and are indispensable [105]. To lower the production investment of single-atom catalysts, Zu et al. synthesized kilogram-scale Sn^δ+^ on N-doped graphene (SnN_2_C_2_) using a rapid freeze-vacuum drying-calcination procedure, and its electroreduction activity basically remained inactive even after 200 h (Figure 5d) [106]. Sulfur and oxygen atoms have been viewed as ideal alternatives for coordinated N in the metal-Nx center because of their lower electronegativity than N, having the ability to dramatically change the active center’s local electronic densities [107]. Wang et al. developed an atomically dispersed N, S co-coordinated Bi site catalyst (Bi-N_3_S/C) shown in Figure 5e, which achieved up to 98.3% FE_CO_ at −0.8 V vs. RHE, which is significantly better than Bi-SAs-N/C, and this demonstrated that the introduction of S can greatly lower the energy barrier of the intermediates of the rate-determining step [108]. Sometimes, other strategies are employed in cooperation with heteroatomic coordination to increase a catalyst’s activity. In a S/N dual-heteroatom anchored unsaturated Ni site (NiN_2_S-V), the decreased energy barriers for the electroreduction of CO_2_ to CO are a result of both doped S atoms and evolved S vacancies [109]. As shown in Figure 5f, this synergistic interaction was also verified by Chen et al. via developing a tandem catalyst consisting of a single Cu site with N and S co-coordinated and atomically dispersed Cu clusters (Cu-S_1_N_3_/Cu_x_) [110]. Guo et al. demonstrated the difference between prepared SnN_3_O_1_ (CO product) and a classic SnN_4_ (HCOOH product) configuration for CO_2_-to-CO conversion [64]. Meanwhile, Sn-C_2_O_2_F catalysts with distinctive non-nitrogen coordination structures also support the opinion just mentioned according to the selectivity of CO_2_RR products [83]. In addition, the smaller radii of the electron-deficient B make it vulnerable to coordinate with the electron-rich N via electron-sharing effects, such as with the B-doping in B/N co-coordinated atomic Fe-based catalyst (Fe-N_3_B) that also facilitated *COOH formation and inhibited hydrogen generation [82]. Indirectly coordinated heteroatoms (secondary or higher coordination shells) can also induce local electric fields and change the catalytic behavior of the catalytic center. Chen et al. reported S doping in the second coordination shell of FeN_4_ (FeN_4_S), shown in Figure 5g, and Gu et al. finished the B bridging atomic coordination of NiN_4_ (NiN_4_B_2_) shown in Figure 5h, both inducing an enhanced CO_2_RR performance relative to pristine FeN_4_ as well as NiN_4_, respectively [111,112]. P, in the N group, as a classic electron donor, through integrating into carbon substrates, can alter the electronic structure of the metal site via its 3p lone pair electrons. In Sun et al.’s work on Fe-SAC/NPC (FeN_4_O_1_-P), by fitting FT-EXAFS spectra in R space, each Fe atom was separated by four Ns and one O, while a single P in high-coordination shells (n ≥ 3) was merged mainly as P-C bonds into the N-doped carbon (Figure 5i), and the P of the third coordination shell enhanced the stability of the key *COOH intermediate on Fe, resulting in an outstanding CO_2_RR performance at a low applied potential [113].

**Figure 5 nanomaterials-13-00309-f005:**
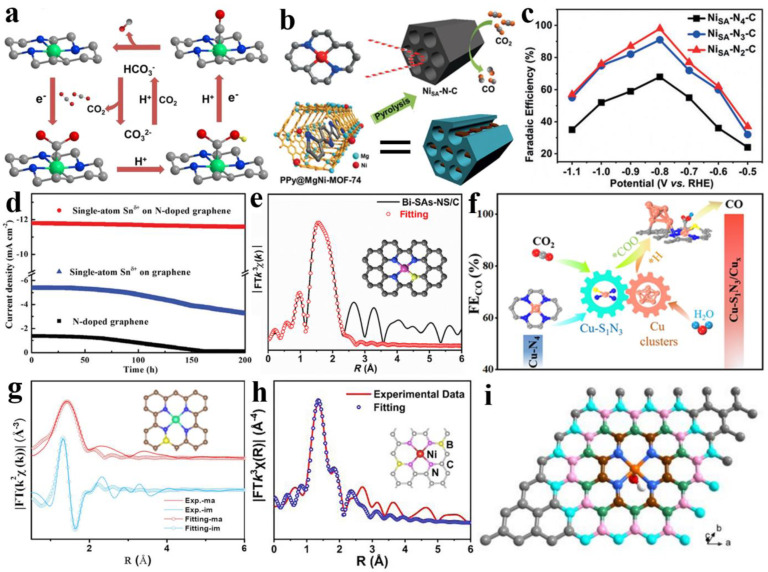
(**a**) Proposed reaction paths of Ni SAs/N-C for CO_2_RR. Reprinted with permission from ref. [102]. Copyright: (2017) American Chemical Society. (**b**) Schematic of preparation of the NiSA-N_x_-C catalysts. (**c**) FE_CO_. Reprinted with permission from ref. [103]. Copyright: (2019) Wiley-VCH Verlag GmbH & Co. KGaA, Weinheim. (**d**) Chronoamperometry test of single-atom Sn^δ+^ on N-doped graphene at the potentials of −1.6 V versus SCE. Reprinted with permission from ref. [106]. Copyright: (2019) WILEY-VCH Verlag GmbH & Co. KGaA, Weinheim. (**e**) The EXAFS fitting for Bi-SAs-NS/C. Reprinted with permission from ref. [108]. Copyright: (2021) Tsinghua University Press and Springer-Verlag GmbH Germany, part of Springer Nature. (**f**) Schematic of Cu-S_1_N_3_/Cu for CO_2_RR. Reprinted with permission from ref. [110]. Copyright: (2021) Wiley-VCH GmbH. (**g**) The fitting EXAFS spectra of Fe_1_-NSC and fitting model. Reprinted with permission from ref. [111]. Copyright: (2022) Wiley-VCH GmbH. (**h**) The FT-EXAFS spectrum for Ni-SAs@BNC. Reprinted with permission from ref. [112]. Copyright: (2022) Elsevier Ltd. (**i**) Different local structures of FeN_4_O-graphene: the second (wine, FeN_4_OP_2_), third (olive, FeN_4_OCP_3_), fourth (pink, FeN_4_OCP_4_), and fifth (cyan, FeN_4_OCP_5_) coordination shell of Fe center. Reprinted with permission from ref. [113]. Copyright: (2021) Wiley-VCH GmbH.

In short, heteroatom coordination has a greater influence on CO_2_RR conversion compared with lower coordination strategies and can even completely change the obtained product, e.g., SnN_4_, a classical formic-acid-producing catalyst, transforms into CO after the introduction of a coordinated O or F. Further enhancement of the understanding of the mechanism of heteroatom action can expand the choice of products.

### 5.3. Axial Heteroatom Coordination Structure

Heteroatoms in the conductive carbon supports can function as bridges to axially couple to MN_4_ sites, while isolated heteroatoms can also build axial coordination patterns in the form of unsaturated coordination (dangling atoms). The various heteroatoms can directly coordinate with the single atomic M center in the symmetric MN_4_ to construct an asymmetric axial coordination structure of MN_4_X_1_ (X = N, O, S, Cl, F), which is essentially identical to the function of the lateral structural configuration, i.e., to obtain a more suitable electron structure and performance than pristine MN_4_.

Zhang et al. designed a graphene-confined atomic dispersed FeN_5_ catalytic site in Figure 6a–c, in which the electron density of the Fe 3d orbitals is further consumed through axial pyrrolic N, thus weakening the Fe-CO π back donation, resulting in a fast desorption toward CO and high selectivity (only at −0.46V vs. RHE, up to ~5 mA cm^−2^, FE_CO_ ca. 97.0%) [114]. After the substitution of axial N by O, the corresponding catalytic activity of the obtained Fe_1_N_4_O_1_ was further strengthened (−0.56 V to −0.87 V vs. RHE, up to ~15 mA cm^−2^, FE_CO_ nearly 100%), as shown in Figure 6d–f [115]. Simultaneously, similar satisfactory results were also achieved for NiN_4_O (−0.5 V to −1.1V vs. RHE, up to ~30 mA cm^−2^, FE_CO_ above 90%), which could be responsible for the axial traction effect with an additional axial oxygen atom in Figure 6g–i [116]. Additionally, the one-pot template sacrificial pyrolysis method used by Huang et al. promises the development of a series of metal-N_4_-O catalysts, such as well-defined Ni-N_4_-O, for a variety of catalytic applications [117]. Aiming to attain the targets of high selectivity, high current density, and long-term stability, Wu et al. designed and prepared a CdN_4_S_1_/CN catalyst that is effective for the electrochemical reduction of CO_2_ to CO (−2.1 V to −2.7 V vs. Ag/Ag^+^, FE_CO_ more than 95%) in H-type cells with [Bmim]PF_6_-MeCN cathodic electrolytes and also reaches an outstanding current density of 182.2 mA cm^−2^, which is majorly owing to the reason that Cd can inhibit HER as well as S with a high spin density and charge delocalization can lower the key free energy barrier. Thus, the CdN_4_S_1_ site has a lower free energy barrier for the CO_2_RR (0.27 eV) and higher free energy barriers for the formation of H* for the HER (0.49 eV) (Figure 6j–l) [118]. Furthermore, Huang et al. designed a new Ni-N_5_-C single-atom catalyst with an enzyme-like catalytic active site and then evaluated its performance using a flow cell device, achieving a maximum current density of 1.23 A cm^−2^ at −2.4 V vs. RHE with a 99.6% FE_CO_ and 100 h continuous stable operation [119]. In addition, in some reported works, Cl and F are also used as axial coordination atoms of MN_4_ to improve the electroreduction properties of CO_2_, such as FeN_4_Cl/NC and MnN_4_Cl, and, although both of them can achieve a high selectivity of >90%, they are limited to a narrow lower potential window [56,78]. In another specialized configuration catalyst of SnC_2_O_2_F, the F atom bonded to Sn has a significant capability to inhibit hydrogen evolution [83].

**Figure 6 nanomaterials-13-00309-f006:**
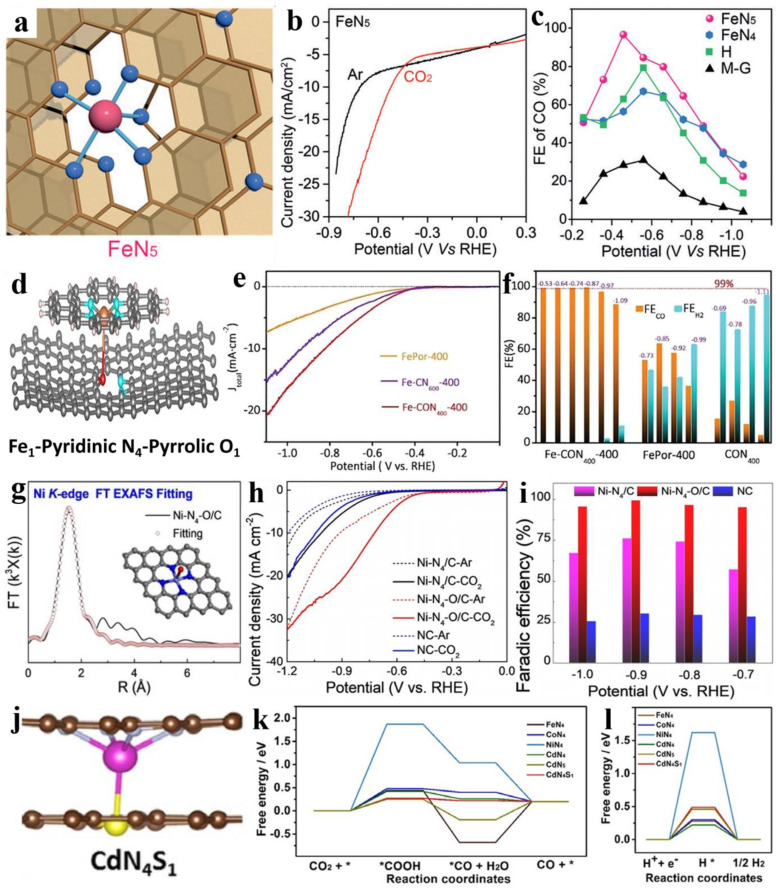
(**a**) The local structure of FeN_5_ catalysts. (**b**) The linear sweep voltammetric (LSV) curves in 0.1 M KHCO_3_ solution. (**c**) FE_CO_. Reprinted with permission from ref. [114]. Copyright: (2019) Wiley-VCH Verlag GmbH & Co. KGaA, Weinheim. (**d**) Optimized configurations for Fe_1_-Pyridinic N_4_-Pyrrolic O_1_. (**e**) LSV curves. (**f**) Faradaic efficiency of CO and H_2_. Reprinted with permission from ref. [115]. Copyright: (2021) The Royal Society of Chemistry. (**g**) EXAFS fitting curves of Ni-N_4_-O/C in R space. (**h**) LSV curves. (**i**) FE_CO_. Reprinted with permission from ref. [116]. Copyright: (2020) Wiley-VCH GmbH. (**j**) The side view of the CdN_4_S_1_ models. Gibbs free energy diagrams for CO_2_RR to (**k**) CO and for (**l**) HER with different models. Reprinted with permission from ref. [118]. Copyright: (2021) Wiley-VCH GmbH.

By summarizing the above-mentioned works, it can be found that the activity of the CO_2_RR is gradually enhanced by modifying the axial coordination atoms or optimizing the choice of metal centers, and the axial coordination atoms seem to be more advantageous in inhibiting hydrogen precipitation. Importantly, the choice of electrolyte, such as ionic liquids, also has a significant effect on the improvement of CO_2_RR yield.

### 5.4. Dual-Metal Coordination Structure

Bimetallic atomic sites, i.e., neighboring monoatomic sites, provide more opportunities for the manipulation of the electrical and geometric structures of SACs on the basis of preserving atomic dispersion. Differing from SACs, bimetallic heteroatomic site catalysts, mainly including M_1_N_4_-M_2_N_4_, M_1_M_1_N_x_, M_1_M_2_N_x_, etc., utilize two adjacent metal atoms to achieve a functional association and synergistic effect and, furthermore, through the electronic interactions between them, to efficiently tailor the binding strength of intermediates as well as optimize CO_2_RR activity.

Xie et al. synthesized a neighboring bi-metal atom catalyst (NiN_4_-SnN_4_) with each Ni and S coordinated with four Ns (Figure 7a). The neighboring Ni redistributes the electrons of Sn and results in the reduced free energy barrier of *OCHO and subsequent thermodynamically spontaneous determining step, and by taking double advantage of the synergistic effect of the NiSn diatomic site and the increased utilization, exhibits excellent selectivity for the generation of formate (86.1% at −0.82 V_RHE_) compared with Sn-SAC (70.4% at −0.82 V_RHE_) as well as Ni-SAC (<1 %), as shown in Figure 7b [120]. Hao et al. designed a non-bridged Ni dual-atom catalyst (Ni_2_N_6_) with a local structure of N_3_-Ni-Ni-N_3_ for CO_2_ to CO (Figure 7c), and in situ environmental STEM captured the Ostwald ripening process of the Ni species generating the bulk phase and the subsequent atomization dynamics process of the transformation into Ni diatomic sites (Figure 7d). The actual active sites for the fast CO_2_RR kinetics and the slower HER kinetics are Ni diatomic sites (Ni_2_N_6_OH) induced via hydroxyl adsorption, according to in situ XAFS studies and theory calculations (Figure 7e). [84] In addition, there is a similar active site on the O-bridge adsorption structure (O-Ni_2_N_6_) of the 2N-bridged Ni_2_N_6_ site for the dynamic catalysis of CO_2_ [85]. Moreover, several other homoatomic binuclear site catalysts with the special local environment and a non-bridged Ni_2_N_4_C_2_ and Pd_2_ dual-atom site with a PdN_2_O_2_ coordination structure have shown superior, nearly 100% CO selectivity [86,121]. 

**Figure 7 nanomaterials-13-00309-f007:**
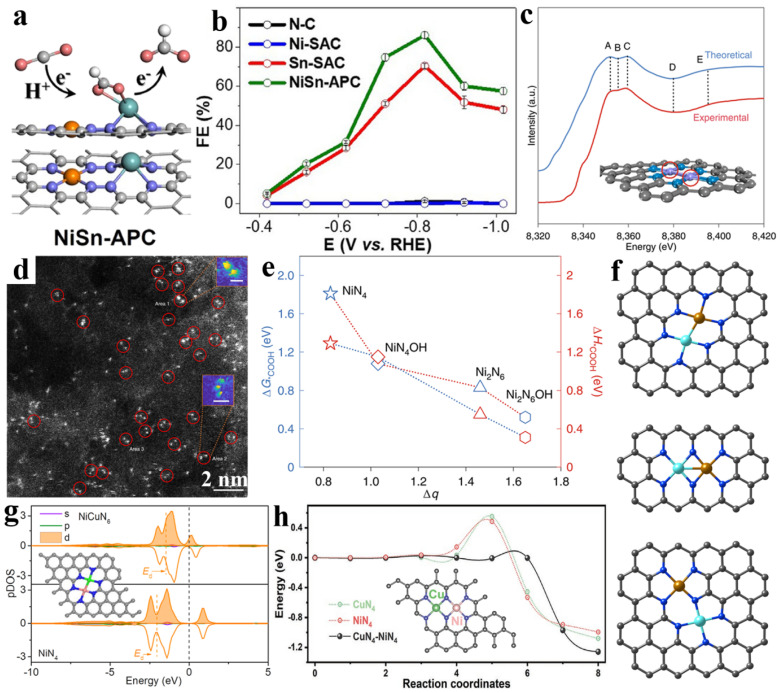
(**a**) Scheme illustration for CO_2_RR of NiSn-APC. (**b**) Faradaic efficiencies for formate. Reprinted with permission from ref. [120]. Copyright: (2020) Wiley-VCH GmbH. (**c**) A comparison between the experimental and the simulated XANES spectra for Ni_2_NC. (**d**) The HAADF-STEM images (inset: the Ni-Ni atoms pair. Area 1, 0.25 nm; area 2, 0.5 nm). (**e**) The relationship between Bader charges (Δq) and Gibbs free energy/Helmholtz free energy for *COOH formation on different sites. Reprinted with permission from ref. [84]. Copyright: (2022) Nature Publishing Group. (**f**) Atomic structure of different dual-metal sites on graphene layer. Reprinted with permission from ref. [122]. Copyright: (2022) Wiley-VCH GmbH. (**g**) Partial density of states (pDOS) of Ni in NiN_4_ and NiCuN_6_. Reprinted with permission from ref. [123]. Copyright: (2022) American Chemical Society. (**h**) Climbing image nudged elastic band (CI-NEB) calculated kinetic reaction processes. Reprinted with permission from ref. [124]. Copyright: (2022) Wiley-VCH GmbH.

To date, some works about the neighboring heterometal M_1_M_2_N_x_ sites (M_1_/M_2_: Fe, Co, Ni, Cu, Zn, etc.) catalysts have been published. Li et al. designed several possible bimetallic site configurations with DFT computational simulations, i.e., non-bridged (Fe-Ni)N_6_, 2N-bridged (Fe-Ni)N_6,_ and 1N-bridged (Fe-Ni)N_7_ sites (Figure 7f). The optimal CO_2_RR activity and selectivity were identified to be at the 2N-bridged (Fe-Ni)N_6_ sites, which can account for these two metal sites’ synergistic interaction that could simultaneously promote the adsorption of *COOH and the desorption of *CO while inhibiting the HER [122]. More in-depth insights can be provided through an electronic structure analysis: in the NiFe heteroatom sites, Fe acts as the catalytic center with a higher oxidation state resulting from its orbital coupling with Ni, which facilitates CO_2_ activation and does not change the binding strength to *COOH but the weakened binding strength with *CO intermediates, and accelerates CO desorption, thus increasing the catalytic CO_2_RR activity [125]. Jiao et al. concluded that in non-bonding Fe_1_-Ni_1_-N-C, the Fe atomic site can also be highly activated by the adjacent single-atom Ni through non-bonding interactions between them to act as the active center [126]. Meanwhile, Ren et al. believe that the key to the CO_2_RR is the collaborative coordination of CO at non-bridged diatomic NiFeN_6_ sites fixed on nitrogenated carbon [127]. 

Furthermore, two types of (Cu-Ni)N_6_ for non-bridged Ni/Cu-N-C (Figure 7g) and 2N-bridged CuNi-DSA/CNF (inset of Figure 7h) have been reported: for non-bridged Ni/Cu-N-C, the introduction of the secondary Cu metal accelerates the rate-determining step of *COOH formation by transferring the Ni 3d orbital energy to the Fermi energy level; for 2N-bridged CuNi-DSA/CNF, electronegativity shifts between Cu and Ni, causing strong electronic interactions and the offset effect, which can efficiently optimize the adsorption strength of *COOH intermediates. The activation energy of *COOH creation on CuN_4_-NiN_4_ is 0.08 eV, as shown in Figure 7h, which is much lower than that on CuN_4_ and NiN_4_ (0.58 eV and 0.48 eV), proving that the CuNi dual-atom sites do actually facilitate the kinetic process of *COOH production [123,124,128]. In addition, co-constructed M_1_M_2_N_x_ composed of other metals, e.g., the ZIF-8 containing Fe/Cu sources derived a Cu-Fe diatomic site catalyst that coordinated with the N and was doped in the carbon matrix (non-bridged (Cu-Fe)N_6_) [65,67]; direct pyrolysis of a combination of carbon black, metal salt, and a urea nitrogen source yielded a Co-Cu diatomic site catalyst (2N-bridged (Co-Cu)N_6_) [129]; a mixture of chitosan and Ni/Zn metal salts derived atomically dispersed Ni-Zn bimetal sites coordinated on N-doped carbon (non-bridged (Ni-Zn)N_6_) [130]; and a precursor containing Zn/Co nitrate, urea, and activated carbon black derived diatomic Zn-Co monomers supported on nitrogen-doped carbon (2N-bridged (Zn-Co)N_6_) [131], have also shown better CO_2_ electroreduction performances. 

The analysis of the above different strategies for improving the CO_2_RR reveals that each strategy has its own unique advantages, and the combined use of these strategies may achieve unexpected results. It is important to note that the final evaluation index of almost all the strategies focuses on the pair energy barrier and adsorption energy changes due to electronic interactions but ignores factors such as hydrogen bonding, solvation effects, and spatial structure that are often simplified or ignored in theoretical analyses. In addition, mesoscopic factors, such as the modulation of the solid-liquid-gas interface, should also be taken into account.

### 5.5. Asymmetric Atom Sites of Organic/Metal-Based Supports for CO_2_RR

Except for carbon-based ones, organic/metal-based supports are also capable of loading the above-mentioned asymmetric coordination atom sites. Organic crystalline materials, including MOFs, covalent organic frameworks (COFs), metal-organic complexes, etc., have explicit structures and tuned metal sites while retaining a major problem of poor electrical conductivity that cannot be ignored.

To overcome the conductivity of a crystallite, one strategy is to enhance conductivity with itself, e.g., conductive MOFs with conjugated organic ligands, and the other is to integrate it with other better conductive materials. Zhang et al. provided coordination-environment-dependent single-site-Cu-O_4_ conductive MOFs (Cu-DBC), and, by comparing the Cu-N_4_ sites of two other COFs (Cu-TTCOF and Cu-PPCOF), it can be found that the Cu-O_4_ site with lower free energy barrier has better CO_2_ to CH_4_ activity (Faradaic efficiency of CH_4_ up to ~80% at −0.9 V vs. RHE) than the CuN_4_ sites (Figure 8a) [57]. Another conductive COF (PcCu-TFPN) with a high electron density and isolated phthalocyanine CuN_4_ sites is more appropriate for the critical step of the C-C coupling of *CH_3_ intermediates with CO_2_ to generate acetates (Faradaic efficiency of 90.3(2)% at −0.8 V vs. RHE), as shown in Figure 8b [132]. To more precisely elucidate the essential contribution of C-C coupling to the CO_2_ deep reduction of C_2+_ products, Shao et al. designed two MOFs with various nodes, and, compared with BIF-104 with isolated Cu sites as a reference, BIF-102 for the Cl^-^ bridging of a dimer Cu unit can provide a higher C_2_H_4_ Faraday efficiency as high as ~11.3%, while the adjacent Cu atom acts as a modulator, changing the reaction barrier, promoting C-C coupling, and providing a reaction pathway for generating C_2+_ products that differ from isolated monomers, which shows that there exists a close correlation between the active-centered coordination structure and CO_2_RR selectivity (Figure 8c) [133]. Zhu et al. hold a comparable view that in robust π–π stacking MOFs (CuBtz) with pyrazolate-bridged dicopper(I) sites, the uncoordinated N atoms act as proton relays and co-catalyze with neighboring Cu sites to promote *CO hydrogenation and the C-C coupling for the highly selective electroreduction of CO_2_ to C_2+_ products (Figure 8d) [134]. Instead, Lin et al. anchored cobalt phthalocyanine with CoN_4_ sites (CoPc) in the FeN_4_ sites of graphene (Fe-N-C) for synergistic catalytic CO_2_ electroreduction, and the received CoPc@Fe-N-C showed a broader potential window and larger CO current density beyond Fe-N-C [135]. Pan et al. used pre-prepared hollow carbon spheres as supports to successfully construct CoN_5_ asymmetric sites through the coordination of Co in CoPC and N-doped carbon spheres (Figure 8e), obtaining a high selectivity of FE_CO_ of over 90% for the CO_2_RR across a broad potential range, from −0.57 V to −0.88 V vs. RHE (Figure 8f) [136]. A similar strategy was successfully employed by Wang et al. to immobilize planar Co^II^-2,3-naphthalocyanine (NapCo) on graphitic sulfoxide (SO) or carboxyl (COO)-doped graphene (NapCo@graphitic SO; NapCo@COO), i.e., where axial coordination between CoN_4_ in NapCo and O from SO or COO occurred (Figure 8g). As a result, there was good electronic communication between NapCo and graphene in NapCo@graphitic SO compared with NapCo@COO, resulting in a high Faraday efficiency of 97% for CO production, yet no activity for CO_2_ electroreduction was found for the CoN_4_ site of pristine NapCo without graphene [137]. In metal-based supports, Jiao et al. reported Cu_1_^0^-Cu_1_^x+^ pairs stabilized on the Te surface imperfections of Pd_10_Te_3_ alloy nanowires, and, with Cu_1_^x+^ and Cu_1_^0^ separately corresponding to the adsorption of molecular H_2_O and CO_2_, with an almost completely suppressed HER, this named bi-atomic activating bimolecular effect facilitates the evolution of CO_2_ to CO (Figure 8h) [138].

Unlike carbon-based supports, there are still relatively few hetero-coordination atoms loaded on organic/metal-based supports, and the key is to fundamentally address organic substrates’ conductivity, stability, and lessened selectivity, as well as metallic carriers’ undesirable competitive HER, but satisfactory CO_2_RR data have still been found from the limited reports, which inspires the majority of researchers to continue exploring this vast uncharted territory.

## 6. Summary and Outlook

For over a decade, significant advancements have been accomplished in the structural design and fundamental electrocatalytic investigation of SACs as next-generation catalysts with a maximum of nearly 100% atomic utilization [139,140,141]. However, for the CO_2_RR, the simplicity of the conventional MN_4_ structure severely restricts the substantial improvement of its electrocatalytic activity. It is commonly accepted that the SACs’ coordination structures determine the electronic structures and thus affect the intrinsic electrocatalytic activity. It follows that the local/expanded coordination environment of active single atoms in SACs has a combined effect on the stability, catalytic performance, and utilization of the loaded materials [142,143]. As shown in Table 1, asymmetric atom sites, which have recently gained popularity among a wide range of researchers, can provide flexible structures with customizable atomic configuration, local/expanded coordination, and electronic architecture without damaging this principle of high atomic utilization, thereby breaking such innate limitations of MN_4_. Therefore, efforts have been made in recent years to develop more advanced asymmetric atom electrocatalysts for the CO_2_RR compared with SACs.

Furthermore, the attention on the CO_2_RR has extended to the catalytic environment, as the catalytic activity can be modulated by altering the structural parameters of the catalyst (solid)-electrolyte (liquid)-molecular reactant (gas) three-phase interface [144,145,146]. Some advances have been realized in these interfacial modulation studies, wherein it was discovered that controlling the proton supply or CO_2_ supply can effectively inhibit the HER [147,148], electrodes with improved interfacial structures significantly improve selectivity and partial current density at relatively low overpotentials in H-type electrolytic cells [149,150], and membrane reactors and microfluidic reactors (flow cells) can achieve higher current densities [151,152]. Modification of the catalyst-electrolyte interface region also includes the addition of external molecules (or polymers), changes in electrolyte composition and concentration, the design of novel reactor and electrode structures, and a combination of these approaches, and these modification strategies can significantly affect the intrinsic and extrinsic catalytic activity of the catalyst. Furthermore, the design of syngas (H_2_ and CO) catalysts can also give us some guidance, especially by regulating the relative content of different active center species, to achieve the selectivity of H_2_ and CO, which can be used to determine the differences in various activities of metal sites for the CO_2_RR and HER and to obtain general design principles [153,154]. Moreover, some predictions about CO_2_RR electrocatalysts from theoretical calculations also provide guidance for the development of new catalysts, which have previously received less attention, for example, Rh@Au(100) and Rh@Ag(100), which may produce CH_4_ [155], and the Cu_2_N_4_-loaded C_2_N layer, which can form CH_4_ and C_2_H_4_ [156], are expected to perform well.

Recent studies on asymmetric atom sites in the electrocatalysis of the CO_2_RR were reviewed in this paper, with importance placed on the comparison of asymmetric atom sites and traditional SACs for the CO_2_RR, including an examination of the crucial role that coordination structures play in the intrinsic electrocatalytic activity of supports-loaded SACs, including local coordination (metal atom center and first coordination shell) and extended coordination (second and higher coordination shell), as well as the resulting alternation in the adsorption parameters, the decrease in the energy barrier of rate-determining step, and the optimization of the reaction pathway. It can be straightforwardly found that asymmetric atom sites have significant advantages in improving the electrocatalytic properties when performing the CO_2_RR over symmetrical atomic catalysts, including catalytic activity, Faraday efficiency, selectivity, stability, etc. It is worth noting that, for the CO_2_RR, this is a universal rule, but some prominent properties of the symmetric coordination should not be neglected, and thus it is concluded that asymmetric atom catalysts are expected to break the theoretical boundary of SACs [157,158]. Furthermore, we proposed the current urgent challenges to be solved and future expected directions in the field of asymmetric atom sites. 

The accurate management of single atoms’ desired coordination environments on the supports remains a major challenge. After the coordination environment is optimized, the corresponding active monoatom is securely trapped in the surface anchor site via the charge transfer effect, interaction absorption, etc., which could successfully avoid the metal atom movement and agglomeration caused by high surface energy during the construction procedure and reaction system, impairing its catalytic efficacy [159,160,161]. Currently, SACs with outstanding performance have been successfully established using some reported synthetic approaches, including wet chemistry, ALD deposition, and high-temperature pyrolysis. However, metal nanoclusters or even NPs are occasionally doped into the resultant SACs. The conventional wet impregnation methods are easy to implement, low-cost, and have a scalable production, but obtaining highly loaded and precisely regulated single-atom coordinated sites is challenging [162,163,164]. The popular high-temperature pyrolysis method generally involves the high-temperature calcination of a pre-designed precursor, but it often has unpredictable and ill-defined multiple coordination structures simultaneously due to the influence of the pyrolysis temperature, heating rate, inert atmosphere, and other factors [165,166,167]. The ALD method, which has the advantage of high-precision control procedures, also has the non-negligible disadvantages of complex synthesis steps, a handful of substrate selection, high investment, and low productivity [168,169,170]. Not negligibly, a suitable choice of supports species will influence the vacancies and defects on the surface to be better utilized for stabilizing isolated metal atoms. Carbon-based materials have become the main choice of carriers for SACs due to the easy introduction of O, S, and N elements, but a major drawback is the instability of carbon-based materials at high temperatures and high oxygen concentrations [171,172,173,174]. Correspondingly, the field of supports should be expanded, such as organic crystallization, metals, and derived oxides and nitrides, as well as the selection of the best preparation techniques. Practically speaking, further breakthroughs in the field of SACs lie in the updating of low-cost, easily scalable, and environmentally friendly preparation methods [175,176,177,178,179,180]. 

Accurate identification of the coordination environment is essential for the development of SACs. STEM techniques allow direct viewing of individual metal atoms to obtain relatively well-defined coordination structures, but the low Z-contrast of the first/second coordination shell (e.g., C, N, S, etc.) causes considerable trouble in mapping the spatial positions of the near-end light atoms. In addition, XAFS characterization can directly distinguish between metal NPs and atomically dispersed atoms, but the acquired coordination structure is generally an averaged result [178,179,180,181,182,183,184,185]. Currently, a combination of characterization techniques, such as electron energy loss spectroscopy (EDS), X-ray photoelectron spectroscopy (XPS), scanning tunneling microscopy (STM), non-contact atomic force microscopy (AFM), and DFT theoretical modeling, is required to obtain a relatively reliable local structure determination [181,182,183,184,185,186,187,188,189]. With the emergence of various in situ/operando characterization techniques in recent years, it has become feasible to directly monitor catalytic reaction processes, which would be extremely potent if applied in conjunction with advanced theoretical modeling. For example, the conversion of dispersed metal atoms into clusters or NPs during electrolysis has been confirmed [184]. A comprehensive insight into the structure-property connection relies on the expression of isolated single atoms rather than statistically averaging data of all the single atoms in diverse bonding structure distributions. The combination of electron microscopy techniques, spectral analysis, and DFT calculations can, firstly, reflect the nature of a single-atom configuration, then, recognize the dynamic evolution of the active components during the formation of products, and, finally, make the reaction pathway and mechanism gradually clear, which is of major interest for catalyst design [190,191,192,193,194,195].

Large-scale industrial applications require the batch construction of SACs with high activity and selectivity using simple and scalable methods, whereby the ultimate goal is to switch out the current noble metal-based electrocatalysts with more accessible transition metals, which is advantageous for the atomic economy and the advancement of green chemistry [188,189,190,191,192,193,194,195,196,197,198]. A practical screening tool is machine learning, which enables the matching and integration of different elements or groups, model construction, and the specific performance prediction of catalysts, and then obtaining optimal results using high-throughput screening, these methods greatly reduce the investment of time and human, material, and financial resources [199,200,201,202,203]. In addition, the factor of long-term thermodynamic stability also needs to be considered [204,205]. Too strong or too weak interactions in the metal-species-anchored supports systems during catalysis can result in negative Ostwald maturation and the migration and aggregation of active sites, respectively [206,207]. In particular, the high temperatures and reducing environments usually faced in industrial applications make catalysts more vulnerable to poisoning or deactivation [199,207]. Overall, the design of high-quality asymmetric atom electrocatalysts via reasonable coordination modulation remains promising.

## Figures and Tables

**Figure 8 nanomaterials-13-00309-f008:**
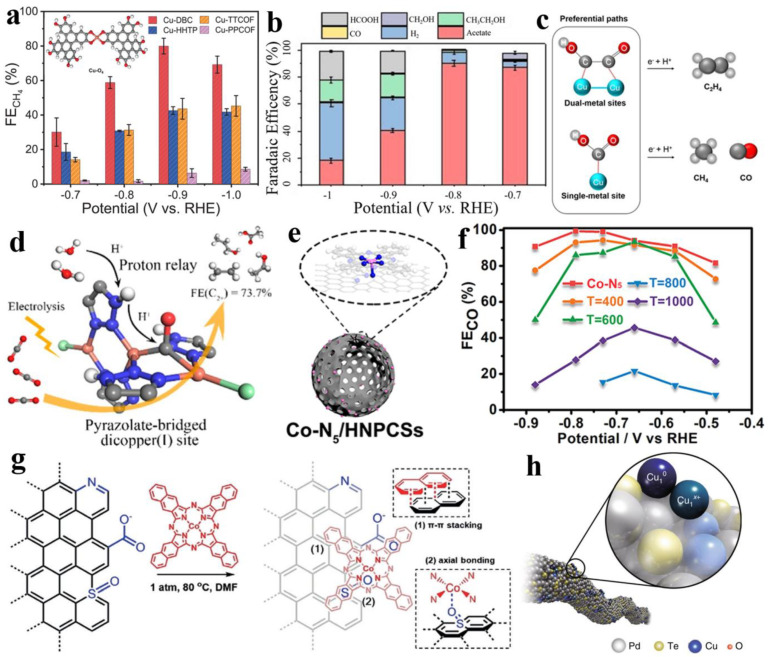
(**a**) Faradaic efficiencies of CH_4_ for crystalline single-site Cu electrocatalysts. Reprinted with permission from ref. [57]. Copyright: (2021) Nature Publishing Group. (**b**) Faradaic efficiencies of different CO_2_RR products for PcCu-TFPN. Reprinted with permission from ref. [132]. Copyright: (2022) Wiley-VCH GmbH. (**c**) The possible preferential reaction pathways of BIF-102NSs and BIF-104NSs. Reprinted with permission from ref. [133]. Copyright: (2021) Wiley-VCH GmbH. (**d**) The possible reaction process of C_2+_ products for CuBtz. Reprinted with permission from ref. [134]. Copyright: (2022) American Chemical Society. (**e**) Schematic illustration of Co-N_5_/HNPCSs. (**f**) FE_CO_. Reprinted with permission from ref. [136]. Copyright: (2018) American Chemical Society. (**g**) Heterogenization of NapCo onto doped-graphene through p-p stacking and coordination with heteroatoms. Reprinted with permission from ref. [137]. Copyright: (2019) Wiley-VCH Verlag GmbH & Co. KGaA, Weinheim. (**h**) Schematic illustration of Cu atom pair anchored on Pd_10_Te_3_ nanowires. Reprinted with permission from ref. [138]. Copyright: (2019) Nature Publishing Group.

**Table 1 nanomaterials-13-00309-t001:** The comparison of CO_2_RR performances of the SACs with different active sites.

Catalyst	Active Site	Electrolyte	Product, FE (%)	Current Density (mA cm^−2^) (E vs. RHE)	Ref.
Fe_1_NC/S_1–_800	FeN_4_	0.5 M KHCO_3_	CO, 82	~2.9 (−0.5 V)	[66]
NiN_4_	NiN_4_	0.5 M KHCO_3_	CO, ~80	~15 (−0.9 V)	[71]
Fe_1_NC/S_1–_1000	FeN_3_	0.5 M KHCO_3_	CO, 96	6.4 (−0.5 V)	[66]
NiN_3_V	NiN_3_	0.5 M KHCO_3_	CO, >90	~60 (−0.9 V)	[71]
Mn–C_3_N_4_/CNT	MnN_3_	0.5 M KHCO_3_	CO, 98.8	14 (−0.55 V)	[101]
Ni SAs/N-C	NiN_3_C_1_	0.5 M KHCO_3_	CO, 71.9	10.48 (−1.0 V)	[102]
NiSA-N_2_-C	NiN_2_C_2_	0.5 M KHCO_3_	CO, ~100	~12 (−0.8 V)	[103]
Single-atom Sn^δ+^ on N-doped graphene	SnN_2_C_2_	0.25 M KHCO_3_	Formate, 74.3	11.7 (−1.6 V_SCE_)	[106]
Sn-NOC	SnN_3_O_1_	0.1 M KHCO_3_	CO, 94	13.9 (−0.7 V)	[64]
Bi-SAs-NS/C	BiN_3_S_1_	0.5 M KHCO_3_	CO, 98.3	~10 (−0.8 V)	[108]
FeN_5_	FeN_4_N_1_	0.1 M KHCO_3_	CO, 97	~5 (−0.46 V)	[114]
Fe-CON_400–_400	FeN_4_O_1_	0.1 M KHCO_3_	CO, ~100	~15 (−0.56~−0.87 V)	[115]
Ni-N_4_-O/C	NiN_4_O_1_	0.5 M KHCO_3_	CO, >90	~30 (−0.5~−0.1.1 V)	[116]
NiSn-APC	NiN_4_-SnN_4_	0.5 M KHCO_3_	Formate, 86.1	~22 (−0.82 V)	[120]
ZIF-NC-Ni-Fe	2N-bridged FeNiN_6_	0.1 M KHCO_3_	CO, >93	~22 (−0.3~−1.0)	[122]
Ni/Cu-N-C	Non-bridged NiCuN_6_	0.5 M KHCO_3_	CO, 97.7	~13.7 (−0.6 V)	[123]

## Data Availability

No new data were created or analyzed in this study. Data sharing is not applicable to this article.

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
