# Peer review of "Asymmetric Coordination Environment Engineering of Atomic Catalysts for CO2 Reduction"

_nanomaterials, 2023, doi:10.3390/nano13020309_

Round 1
Reviewer 1 Report
The authors described the application of single-atom catalysts (SACs) to electrochemical CO2 reduction (ECO2RR) introducing diverse examples. Especially, they addressed asymmetric atom sites that are different from the symmetric structure. Furthermore, they classified asymmetric coordination into several groups. (low-coordination, lateral heteroatom coordination, axial heteroatom coordination, dual-metal coordination, and organic/metal-based supports). The manuscript is well-categorized and quite worthwhile. However, there are issues to be addressed for publishing.
1. There are some review articles about single-atom catalysts for ECO2RR. However, there are few reports which deal with asymmetric atom sites. In that respect, this manuscript is of particular interest. The review article must be easy to understand for beginners who want to study the related field. However, the background information of this work is insufficient. Therefore, a more detailed explanation of the concept (symmetric and asymmetric & coordination environment etc.) is suggested.
2. This work focused on asymmetric coordination environment on SACs. However, the title of the manuscript is too wide. It is suggested to rename the appropriate title.
3. The authors well-categorized asymmetric atom sites into several groups. However, the characteristics of each group (4.1 - 4.5) need to be more specifically described to clearly distinguish how each group is different. It is proposed to describe the concept of each group in detail.
4. This review is mainly focused on what type of asymmetric coordination structure. In SACs for catalytic reaction field, it is important to identify which site is the active site for the catalytic reaction. However, there is a lack of information about the characterization in this manuscript. Therefore, it is suggested to supplement the new part for characterization. In addition, it needs to inform what is important to analyze an asymmetric single-atom site compared to a symmetric single-atom site.
Author Response
please see the attach file.

Reviewer 2 Report
This manuscript has been written well based on the structures and properties of single atom catalysts. Especially DFT calculation suggest exact and excellent approaching method for the prediction of the energy of the CO2 reduction reaction. The functions of SAC are giving appropriate energy for CO2 reduction reaction and control of the transferring electrons for the selective product depending on the size and constituents of SAC. But authors did not describe on the size and different energy level of the donating electrons for the selective production of CO2 . So they should add more description on it.
Author Response
please see the attach file.

Round 2
Reviewer 1 Report
I am satisfied with the responses of the authors to my comments and relevant revision. Now I recommend its acceptance in the current form.